# The Selection of Cutting Speed to Prevent Deterioration of the Surface in Internal Turning of C45 Steel by Small-Diameter Boring Bars

Tomáš Vopát *, Marcel Kuruc, Boris Pätoprstý, Marek Vozár, František Jurina, Barbora Bočáková, Jozef Peterka, Augustín Görög and Róbert Straka

Institute of Production Technologies, Faculty of Materials Science and Technology in Trnava, Slovak University of Technology in Bratislava, 917 24 Trnava, Slovakia; marcel.kuruc@stuba.sk (M.K.); boris.patoprsty@stuba.sk (B.P.); marek.vozar@stuba.sk (M.V.); frantisek.jurina@stuba.sk (F.J.); barbora.bocakova@stuba.sk (B.B.); jozef.peterka@stuba.sk (J.P.); augustin.gorog@stuba.sk (A.G.); robert.straka@stuba.sk (R.S.)
* Correspondence: tomas.vopat@stuba.sk

**Abstract:** The turning of small-diameter deep holes is usually critical when the process of machining is unstable and the use of a special boring bar is often necessary. This paper is focused on the influence of cutting speed with a combination of cutting conditions such as feed and tool overhang on chatter marks, surface roughness and roundness of machined holes. In the experiment, two types of tool material for indexable boring bars were used, namely cemented carbide and steel. These are a group of boring bars used for the internal turning of holes of small diameters with indexable cutting inserts. Monolithic carbide boring bars are already used for internal turning of holes of even smaller diameters. Uncoated turning inserts made of cermet were used. The cutting tests were performed on the DMG CTX alpha 500 turning center. In the case of the steel boring bar, decreasing the cutting speed really led to an increase in the quality of the surface roughness and reduced the formation of chatter marks and large chatter marks. The cemented carbide boring bar also followed a similar trend, but it should be noted that the overall effect was not so great. This means that increasing the cutting speed makes the cutting process less stable and, vice versa, lower values of cutting speed reduce the formation of chatter marks and the related deterioration of the surface quality. The occurrence of chatter is directly related to the increase in the surface roughness parameters Ra and Rz of the machined surface. It can be stated that the dependence of roundness deviations on cutting speed values has a similar character to the results of the measured surface roughness values. Therefore, if the cutting speed is increased, it will make the cutting process less stable; this is also indirectly reflected in larger roundness deviations. However, it is necessary to state that this phenomenon can be observed in turning holes with small diameters using the steel boring bar, where the unstable cutting conditions materialized in the form of chatter marks.

**Keywords:** deep hole turning; boring bar; cutting parameters; surface roughness; roundness; cutting speed; tool overhang

## 1. Introduction

Comprehensive reviews on the topic of reducing unstable conditions in machining processes have already been conducted by multiple authors. Internal hole turning is a machining technology containing several specifics in comparison to other internal hole production technologies such as drilling [1]. In most cases, there is information on the production of deep holes; individual publications focus on the weakest point of this technology, which is the low rigidity of the cutting tool or cutting tool holder. When producing deep holes, there is an unfavorable ratio of the hole depth as well as the length of the holder or tool, L, to the diameter of the hole, D, i.e., L/D. Low stiffness of the tool can

lead to the incidence of vibration during machining, occurrence of dimensional inaccuracy and deterioration of the surface roughness. In such cases, researchers focus on various possibilities of increasing the rigidity of tool holders or reducing vibration through various design-technological modifications [2].

When the dynamic stiffness of the boring bar is increased significantly, it leads to a notable increase in the chatter-free cutting depths [3]. The influence of composite boring bars on the chatter initiation when using carbon steel as the turning medium was investigated. The corner radius of the cutting insert was found to be affecting the vibrations, as the cutting insert with a corner radius of 0.4 mm reduced the chatter by more than 50%. Tool wear and machined surface roughness were also observed to be lower in the experiments with reduced vibration [4]. More turning experiments by the same authors were performed, investigating stainless steel borings using two different cutting inserts with corner radius sizes of 0.8 and 0.4 mm. An artificial neural network was used to predict the surface roughness, tool wear and workpiece vibration. An approach like this can be used to select proper cutting tools and parameters to avoid vibration [5].

The pursuit of improved machining processes with the goal of achieving superior surface quality and increased efficiency has driven researchers to explore innovative cutting techniques. In this context, small-diameter boring bars, despite their inherent low stability, are frequently utilized in certain applications where their use cannot be readily replaced. Consequently, researchers have directed their efforts towards improving machining parameters to enhance the process stability and optimize the turning performance when employing these small-diameter boring bars [6–8]. The use of small-diameter boring bars presents a complex challenge, as their reduced tool diameters necessarily result in lower rigidity and increased susceptibility to vibrations during machining. These factors can lead to decreased cutting performance, reduced surface quality and accelerated tool wear, limiting their application in traditional turning.

In recent studies, researchers have focused on addressing the instability associated with small-diameter boring bars by investigating novel cutting-edge designs, advanced tool materials and optimized machining parameters. Understanding the influence of cutting speed, feed rate, depth of cut and other process variables becomes crucial in mitigating the adverse effects of low stability during turning operations. By tailoring these parameters, it becomes possible to achieve improved process stability and, subsequently, enhanced machining efficiency and surface quality [2].

Several studies have sought to optimize machining parameters when utilizing small-diameter boring bars in various turning applications. An adaptive sliding mode control approach can suppress the chattering phenomenon in the boring process in the presence of model uncertainties and unmodeled dynamics; it acts effectively in improving the process stability. A higher material removal rate can be obtained without stability loss [9].

The research in [10] studied the effect of cutting speed and feed on the roundness and hardness when turning stainless steel SUS 303 material.

Looking at the research regarding internal turning and dynamic stability or tuned mass dampers conducted up until now reveals that much research is focused on tools with diameters of 16 mm or more, as shown in Figure 1. This graph was plotted as an overview of individual research focusing on the field of machining with boring bars [2,11–23].

For this reason, the authors of the article focused their interest (black rectangle in the graph) on the small-diameter boring bars with indexable cutting inserts where new findings are expected in this field.

Selecting appropriate cutting conditions for machining with long overhang tools is also an important aspect to consider with regard to the stability of the machining process, especially the parameters of depth of cut and spindle speed [24]. The tool overhang that is more than 4D/5D is often considered a critical value [14,25–27]. Machining with a cutting tool (with respect to the capabilities of the cutting tool) with a tool overhang that is larger than 5D is often problematic. The authors demonstrate that it is important to maintain the ratio of tool overhang to achieve stable cutting conditions and also highlight the effect of

the cutting speed. Therefore, the ratio of L/D in the experiment was selected above and below the critical value (4–5), namely 3.75 and 6. In some cases, where the tool overhang is already critical, the dimensions of the internal holes require the use of boring bars. In these cases, it is necessary to use special types of boring bars or to set suitable cutting parameters. From the experiment, it is noticeable that chatter marks can be prevented by setting the cutting speed correctly.

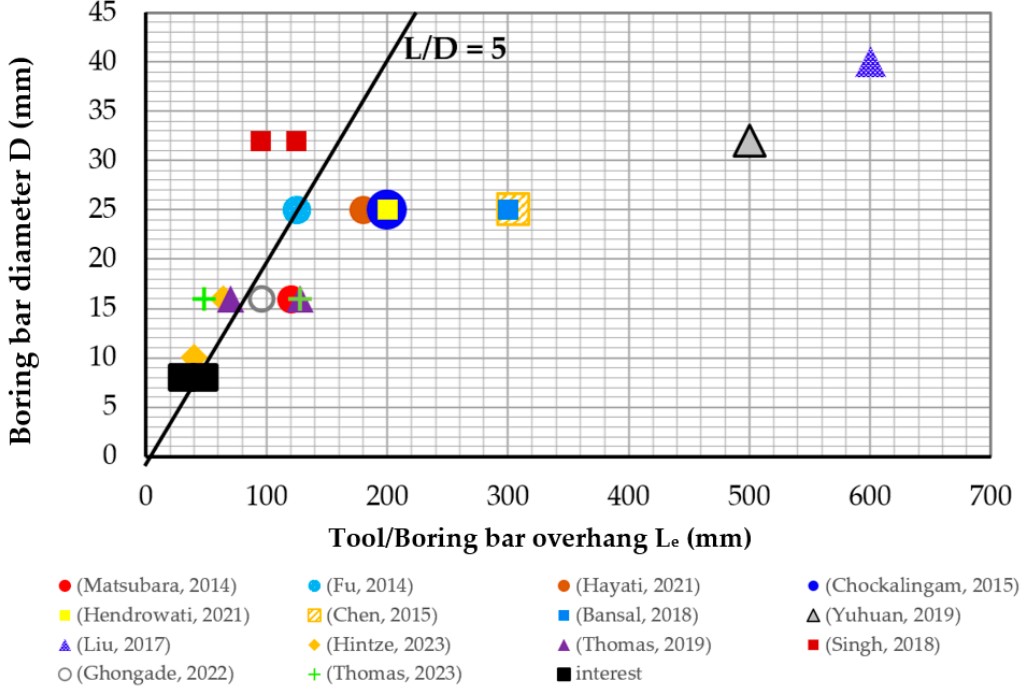

**Figure 1.** Research dealing with machining with long tool overhang.

This scientific paper aims to contribute to this ongoing research by exploring the effects of cutting speed (with respect to the cutting conditions) on the quality of the surface in the internal turning of C45 medium carbon steel, providing valuable insights for engineers and researchers in their efforts to enhance productivity and precision in challenging machining applications.

## 2. Materials and Methods

In the experiment, the influence of cutting speed on chatter marks, surface roughness and roundness was investigated. Moreover, the effect of cutting speed combined with different machining conditions such as feed, tool overhang and tool material of the boring bar (steel and cemented carbide) was also identified.

### 2.1. Selected Cutting Tools and Workpiece Material

The uncoated turning inserts made of cermet with a designation of EPMT 050202E-NF2 (inscribed circle with a diameter of 5.56 mm, thickness of 2.38 mm and corner radius of 0.2 mm) were used. This type of substrate was selected concerning the workpiece material (C45) and finishing/semi-finishing strategy. Two types of boring bars, namely S 0608 H SELPL-05 and E 0608 H SELPL-05, were selected for this research. The shape and specification of these boring bars are described in Table 1 and Figure 2. The selected workpiece material was medium carbon steel of ISO C45 (AISI 1045) grade in this experiment.

**Table 1.** The specification of selected boring bars.

| Boring Bar | D (mm) | BD (mm) | WF (mm) | LF (mm) | Material |
|---|---|---|---|---|---|
| S 0608 H SELPL-05 | 8 | 6 | 4.5 | 100 | Steel |
| E 0608 H SELPL-05 | 8 | 6 | 4.5 | 100 | Cemented carbide |

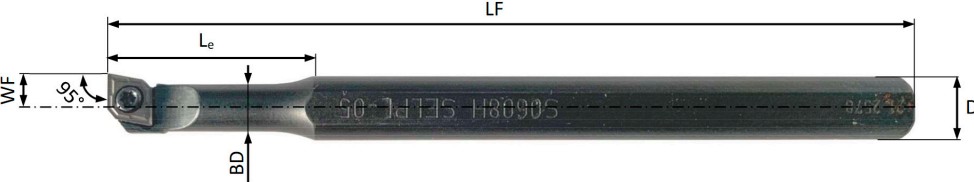

**Figure 2.** Schema of selected boring bar.

*2.2. Cutting Tests*

In the experiment, 32 trials were performed. From cutting parameters, the cutting speed was varied at 4 levels, while the feed was varied at 2 levels. The aim of the experiment is to investigate the effect of cutting speed and the combination of cutting conditions because the main benefit is expected from the preliminary results. For this reason, the feed was varied at 2 levels to change the parameters for the cutting speed. The depth of the cut was kept constant. It is well known from the theory of machining that with increasing values of the depth of cut, the radial force of the tool on the workpiece also increases [28,29]. This is the reason why the authors decided not to change (increase) the values of the depth of cut. Tool overhang lengths were 30 and 48 mm. These selected tool overhang lengths represented the ratio (L/D—tool overhang length/diameter of the boring bar), where tool overhang is smaller than 4D/5D (5 × diameter of the boring bar) for $L_e$ of 30 mm (3.75D) as well as larger than 4D/5D for $L_e$ of 48 mm (6D). As demonstrated in the state-of-the-art research [30], a tool overhang value of more than 4D/5D is often considered critical from the standpoint of chatter initiation in the machining process, which in turn negatively influences the resulting workpiece qualities, such as surface roughness or dimensional accuracy [31]. The cutting parameters used in the experiment are shown in Table 2. The experiment was created particularly for two types of material of the boring bar: steel and cemented carbide.

**Table 2.** Cutting parameters.

| Cutting Parameter | Level 1 | Level 2 | Level 3 | Level 4 |
|---|---|---|---|---|
| Cutting speed $v_c$ (m·min$^{-1}$) (spindle speed (min$^{-1}$)) | 34 (888) | 86 (2245) | 138 (3602) | 190 (4960) |
| Feed f (mm) | 0.02 | 0.10 | | |
| Tool overhang length $L_e$ (mm) | 30 | 48 | | |
| Depth of cut $a_p$ (mm) | 0.10 | | | |

For cutting tests, the DMG CTX alpha 500 turning center was used. The workplace of the turning center with the workpiece can be seen in Figure 3. An oil-in-water emulsion with 5% oil concentration was used as a coolant. From the workpiece material, 32 samples were made and surface roughness parameters Ra and Rz and roundness were measured. The shape of the workpiece was a round rod with an external diameter of 20 mm. Before performing the cutting tests, a hole with a diameter of 12 mm was made by drilling. The drawing of a sample with the explanation of measurements can be seen in Figure 4. After cutting tests, the machined holes were observed in detail by a DinoLite optical digital microscope, and potential chatter marks were evaluated.

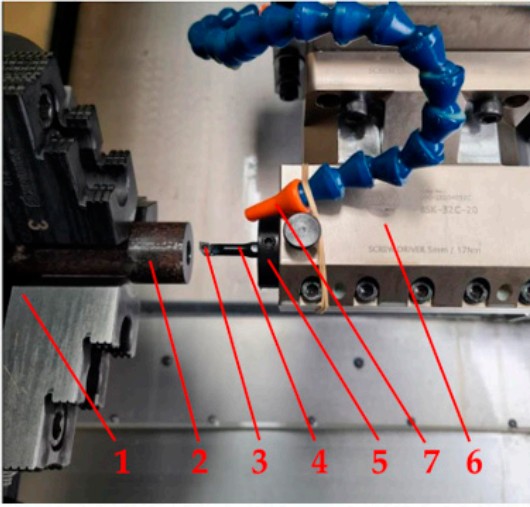

**Figure 3.** Workspace of turning center with workpiece in the cutting tests.

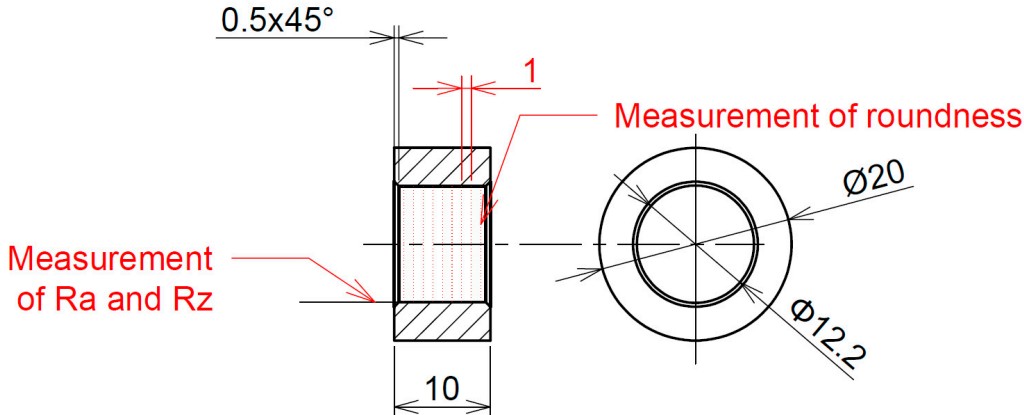

**Figure 4.** Drawing of a sample for surface roughness and roundness measurements.

### 2.3. Measurement of Surface Roughness Parameters Ra, Rz

After cutting tests, the surface roughness parameters Ra and Rz of the turned hole were evaluated. In the experiment, the surface roughness parameters Ra and Rz were measured by the Mitutoyo Surftest SJ-210 surface roughness measuring instrument. The recommended cut-off parameters that were selected for this experiment are shown in Table 3. Ra and Rz parameters were measured three times on the surface of the turned hole for each sample.

**Table 3.** Recommended cut-off parameters for measurement of Ra and Rz.

| Recommended Cut-Off (ISO 4288-1996) | | | |
|---|---|---|---|
| **Non-Periodic Profiles** | | **Cut-Off** | **Sampling Length/Evaluation Length** |
| **Rz (μm)** | **Ra (μm)** | **λc (mm)** | **λc (mm)/L** |
| >0.5–10 | >0.1–2 | 0.8 | 0.8/4 |

### 2.4. Measurement of Roundness

The roundness was measured on a Rondcom 60A high-precision roundness measuring instrument by Accretech (Figure 5). The given accuracy values are a rotation precision of 0.02 μm and a *z*-axis precision of 0.05 μm. Before the measurement, the instrument was calibrated using a standard calibration ball sample. The parameters of the roundness measurement are shown in Table 4. The roundness measurement is performed using a

profilometric sensor that moves over the surface of the sample and records data at surface height. Scanning is carried out around the entire circumference of the measuring point.

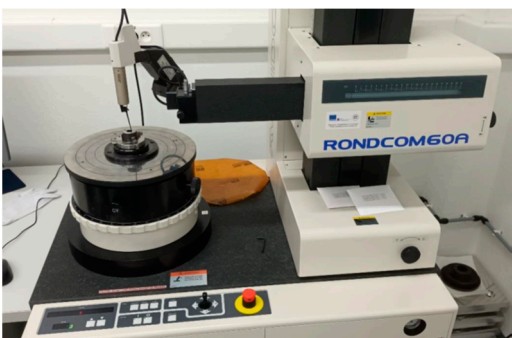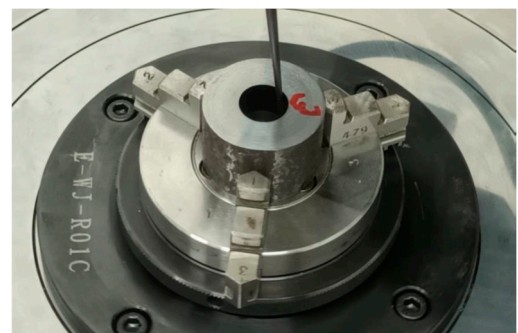

**Figure 5.** Roundness measurement.

**Table 4.** Parameters of roundness measurement.

| | |
|---|---|
| Number of samples | 32 |
| Number of points measured around the circumference | 3600 |
| Number of cuts | 9 |
| Distance between cuts | 1 mm |
| Measuring speed | 3 min$^{-1}$ |
| Evaluation method | MZC |
| Filtering method | GAUS 50 UPR |

## 3. Results and Discussion

As mentioned in the previous section, 32 samples were made and surface roughness parameters and roundness were measured. Examples of the samples with machined surfaces are shown in Figure 6. Machined surfaces of the samples were sorted into three groups according to the observed size of chatter marks: no chatter marks (in Figure 6a,b), chatter marks in Figure 6c,d) and large chatter marks (Figure 6e,f). The chatter marks and large chatter marks are clearly observed as proof of unstable machining conditions.

### 3.1. Surface Roughness

The occurrence of chatter marks is directly related to the increase in surface roughness parameters Ra and Rz on the machined surface. Therefore, the specific chatter marks, such as no chatter marks (marked by a green circle), chatter marks (orange rhombus) and large chatter marks (red triangle), were inserted into the graphs of the average values of the surface roughness parameters Ra and Rz (Figures 7 and 8) as well as roundness (in the Section 3.2.) for better explanation.

As seen from the graphs in Figures 7 and 8, the cemented carbide boring bar is possible for use at a specific range of cutting speeds with a sufficient quality of surface roughness (Ra < 0.8 μm, Rz < 4 μm and no chatter marks) for internal turning with a tool overhang of 6D (6 × diameter of boring bar) in contrast to the steel boring bar. The steel boring bar is only viable to be used for internal turning with a tool overhang of 3.75D or less when a sufficient quality of surface roughness is achieved.

The lowest surface roughness parameters Ra and Rz were reached for a cutting speed of 86 m·min$^{-1}$, a feed of 0.02 mm and a tool overhang length of 30 mm for steel as well as cemented carbide boring bars.

As seen from the graphs (Figure 9), it can be stated that decreasing the cutting speed leads to an improved surface quality of the turned holes. However, it is necessary to realize with this statement that it cannot be applied as a universal phenomenon in machining. In fact, it is necessary to analyze the whole situation and define this phenomenon only for the conditions of internal turning using small-diameter boring bars.

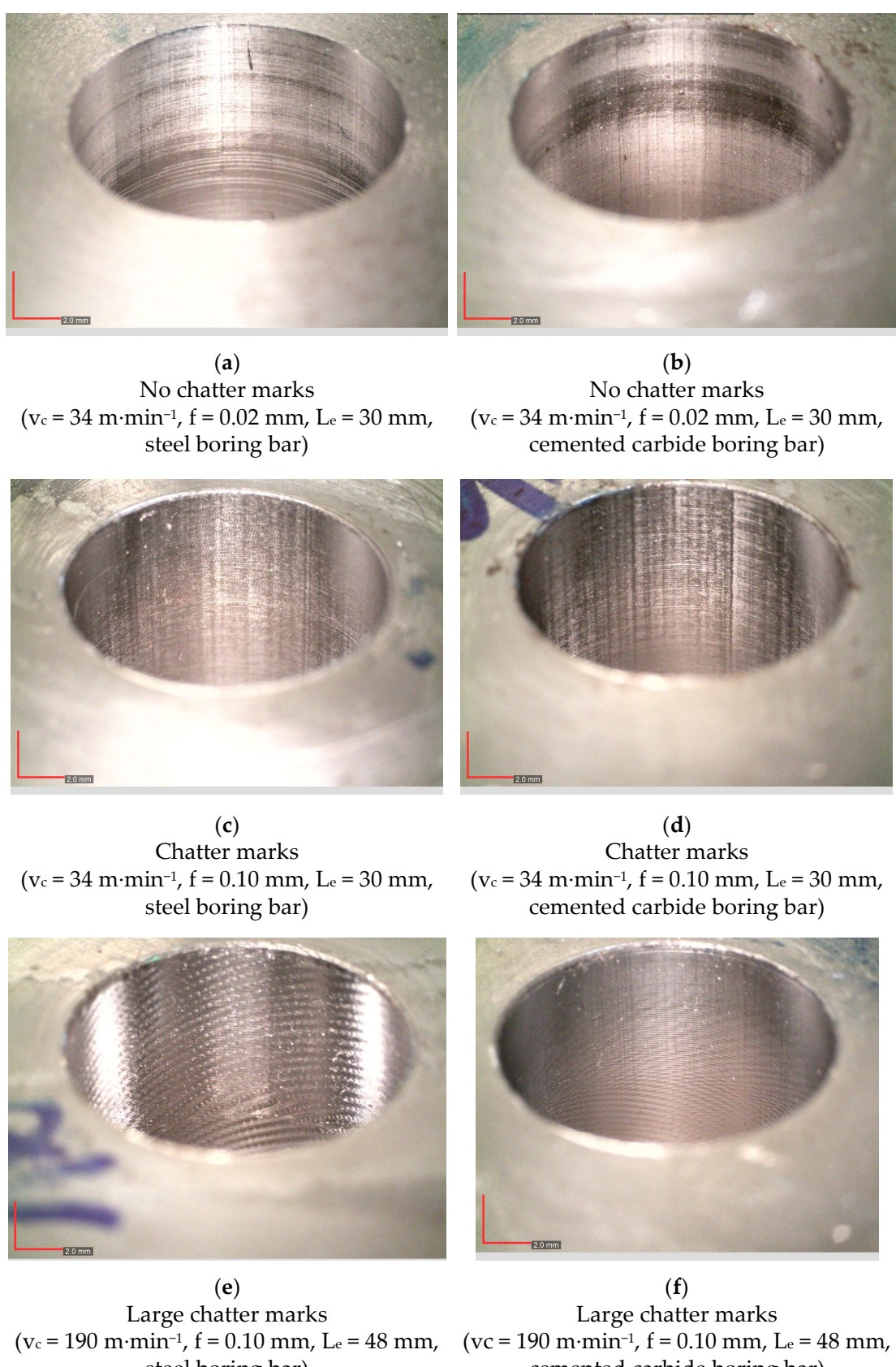

(**a**)
No chatter marks
($v_c$ = 34 m·min$^{-1}$, f = 0.02 mm, $L_e$ = 30 mm,
steel boring bar)

(**b**)
No chatter marks
($v_c$ = 34 m·min$^{-1}$, f = 0.02 mm, $L_e$ = 30 mm,
cemented carbide boring bar)

(**c**)
Chatter marks
($v_c$ = 34 m·min$^{-1}$, f = 0.10 mm, $L_e$ = 30 mm,
steel boring bar)

(**d**)
Chatter marks
($v_c$ = 34 m·min$^{-1}$, f = 0.10 mm, $L_e$ = 30 mm,
cemented carbide boring bar)

(**e**)
Large chatter marks
($v_c$ = 190 m·min$^{-1}$, f = 0.10 mm, $L_e$ = 48 mm,
steel boring bar)

(**f**)
Large chatter marks
($v_c$ = 190 m·min$^{-1}$, f = 0.10 mm, $L_e$ = 48 mm,
cemented carbide boring bar)

**Figure 6.** Samples with machined surfaces where specific kinds of chatter marks are detected: (**a**,**b**) no chatter marks; (**c**,**d**) chatter marks; (**e**,**f**) large chatter marks.

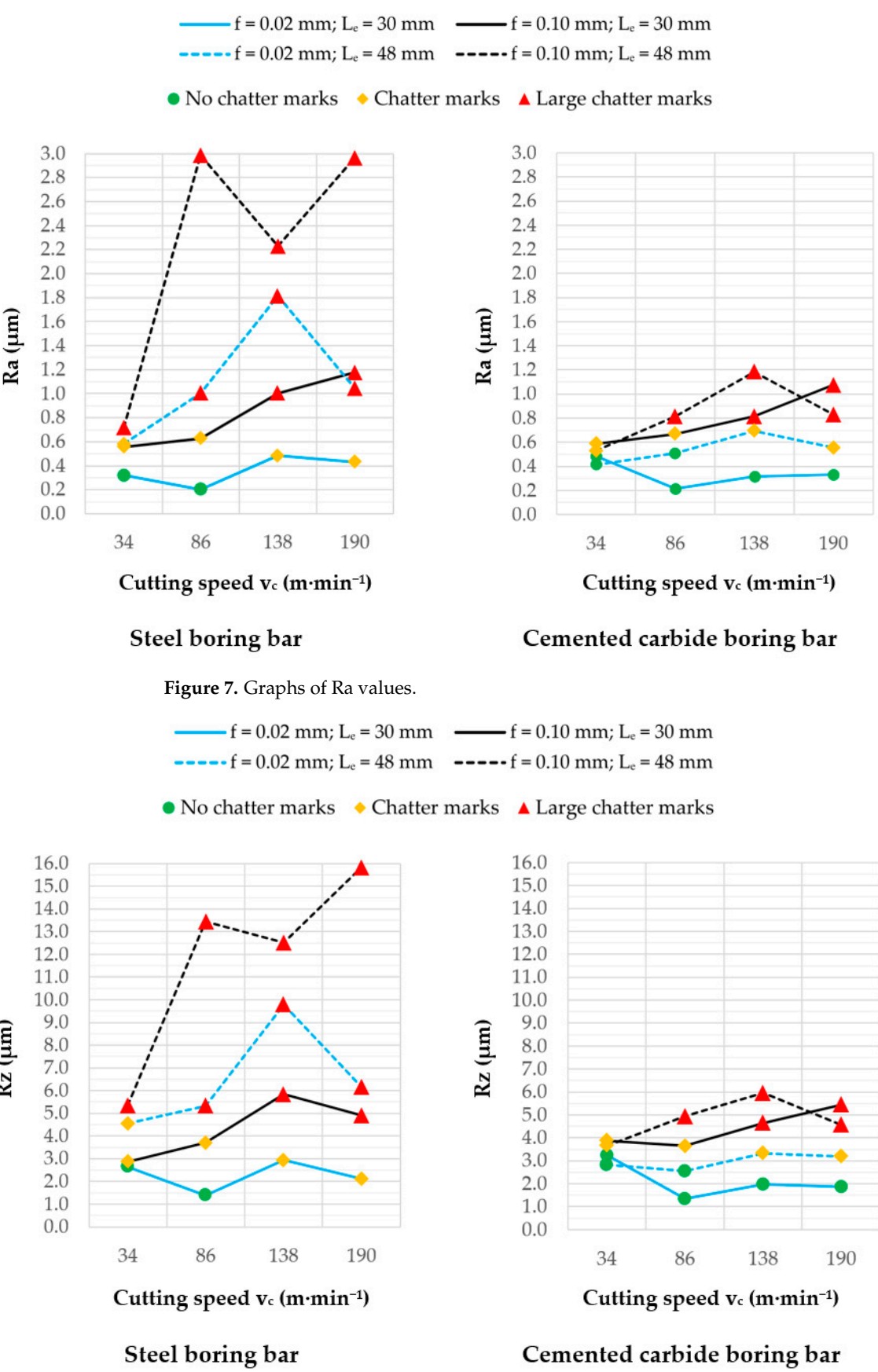

**Figure 7.** Graphs of Ra values.

**Figure 8.** Graphs of Rz values.

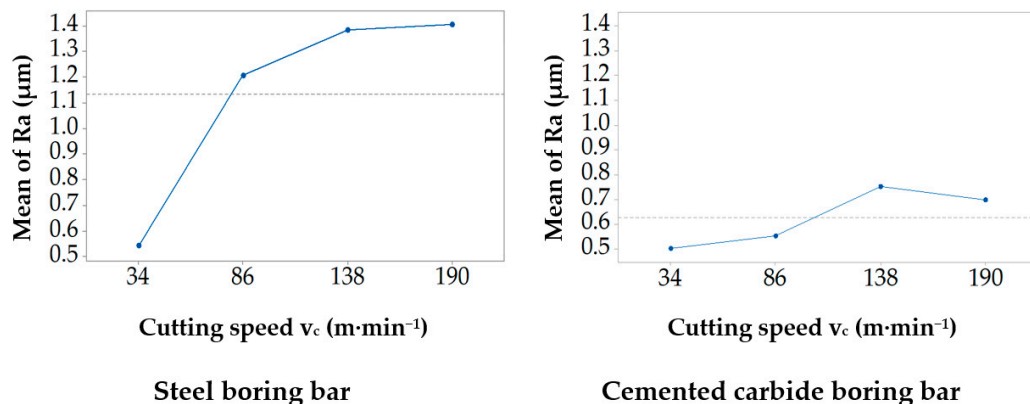

**Figure 9.** Main effect plots for surface roughness parameter Ra (fitted means).

In the case of the steel boring bar, decreasing the cutting speed indeed leads to an increase in the machined surface quality, as illustrated in Figure 9 (overall lower surface roughness parameter Ra), and a reduction in the formation of chatter marks and large chatter marks, as seen from the graphs (Figure 7).

The results provide interesting findings for this area of machining, as they differ from other scientific publications by being complemented by internal turning with small-diameter boring bars. For this reason, when the machining conditions are in between stable and unstable, the effect of the cutting speed is significant, which is different from the statements in [29,32,33] where the cutting speed does not have a significant impact on the surface roughness.

Observing Figure 9, it was found that by reducing the cutting speed, it is possible to achieve a better surface roughness, which is contrary to other scientific publications. In publication [34], surface roughness measurements showed that increasing the cutting speed resulted in lower roughness values.

It is found in [35] that the cutting speed will significantly influence the surface roughness. This publication states that the lower roughness values were reached by increasing the cutting speed.

As mentioned before, the internal turning of small-diameter holes is problematic due to cutting tools with small bar diameters, which do not possess high stiffness. It means that increasing the cutting speed makes the cutting process less stable and lower values of cutting speed reduce the formation of chatter marks and the related deterioration of the surface quality. The authors assume that this would be related to lower values of tangential cutting force at lower cutting speeds. Therefore, further research will aim to prove this statement.

The cemented carbide boring bar also displayed a similar trend, but it should be noted that the overall effect is not as significant. The reduction in chatter mark and large chatter mark formation is also observed in Figure 7. As can be seen from the graph in Figure 9, the dependence curve rises slightly from a cutting speed value of 34 to a value of 86. Then, the dependence curve rises from a cutting speed value of 86 m·min$^{-1}$ to a value of 138 m·min$^{-1}$. After that, the dependence curve only slightly decreases from the value of 138 m·min$^{-1}$ to the value of 190 m·min$^{-1}$, but it needs to be stated that this difference is not significant. For this reason, we can consider that it does not change the overall character of the dependence curve.

In the case of feed, it is necessary to take into account the well-known fact that by imprinting the shape of the nose radius into the machined surface, the surface roughness deteriorates at higher feed values [36–38]. Therefore, it is possible to observe an increase in Ra and Rz values with increasing feed values (from 0.02 to 0.10 mm), as expected. In this case, it is necessary to focus on the formation of chatter marks on the surface of holes. From the graphs in Figures 7 and 8, it is clear that increasing feed values negatively impacted the formation of chatter marks (i.e., from no chatter to chatter marks or large chatter marks).

### 3.2. Roundness

The results of roundness deviations complement the achieved results of surface roughness well. The examples of roundness measurements are shown in Figures 10 and 11. The color roundness profiles measured in one of the nine levels clearly show the difference in the size of roundness deviations (dimension between upper and lower deviation, marked by red arrow) for the limit values of cutting speed (34 and 190 m·min$^{-1}$).

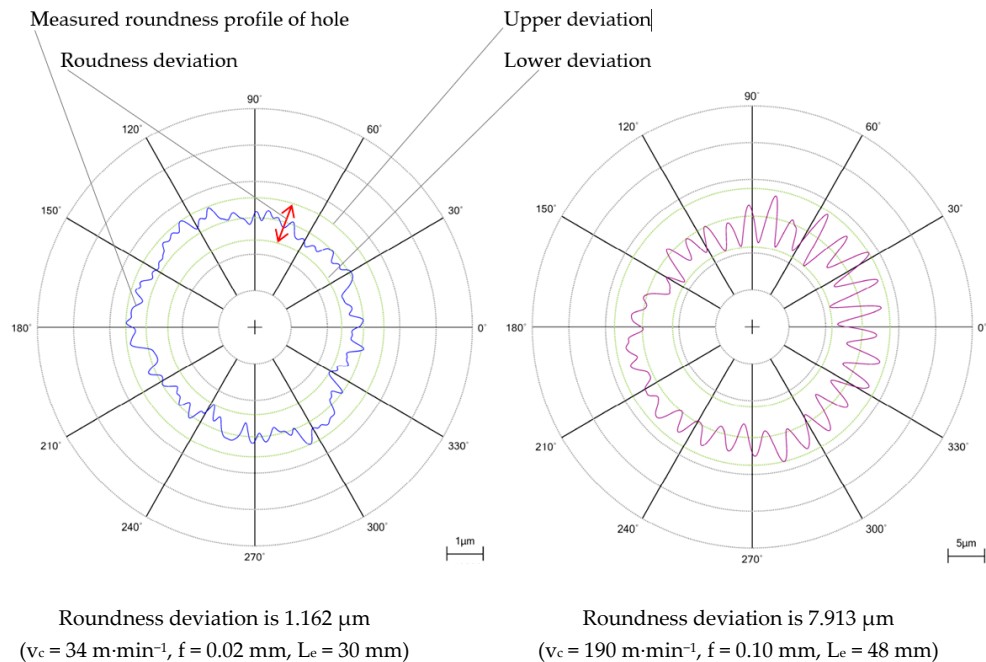

Roundness deviation is 1.162 μm
($v_c$ = 34 m·min$^{-1}$, f = 0.02 mm, $L_e$ = 30 mm)

Roundness deviation is 7.913 μm
($v_c$ = 190 m·min$^{-1}$, f = 0.10 mm, $L_e$ = 48 mm)

**Figure 10.** Examples of the results of roundness measurement for steel boring bars.

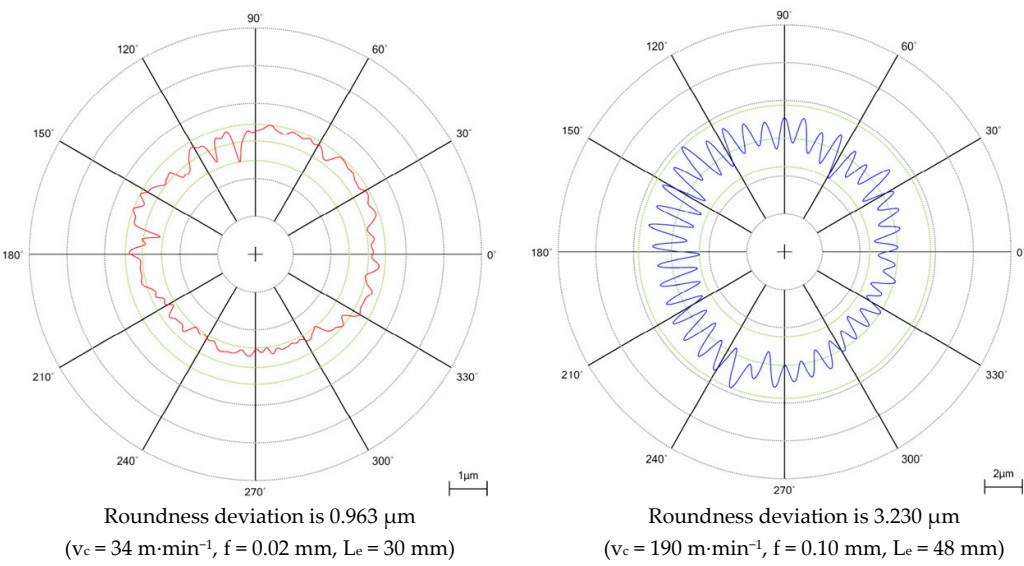

Roundness deviation is 0.963 μm
($v_c$ = 34 m·min$^{-1}$, f = 0.02 mm, $L_e$ = 30 mm)

Roundness deviation is 3.230 μm
($v_c$ = 190 m·min$^{-1}$, f = 0.10 mm, $L_e$ = 48 mm)

**Figure 11.** Examples of the results of roundness measurement for cemented carbide boring bars.

It should be noted that roundness deviations depend on various factors such as the toughness of the machine tool system and many others. For this reason, differences in roundness deviations were compared between samples in the experiment. As seen from the graph (Figure 12), the lowest values of roundness deviation were reached at a cutting speed of 86 m·min$^{-1}$, feed of 0.02 mm and tool overhang length of 30 mm for the steel as well as cemented carbide boring bars. This result is the same as the result from the surface roughness measurement. The samples with machined holes without chatter marks and a

sufficient quality of surface roughness reached roundness deviations that were less than 1.6 μm. The samples with machined holes where only large chatter marks were observed reached roundness deviations of more than 3 μm, which is almost twice as large.

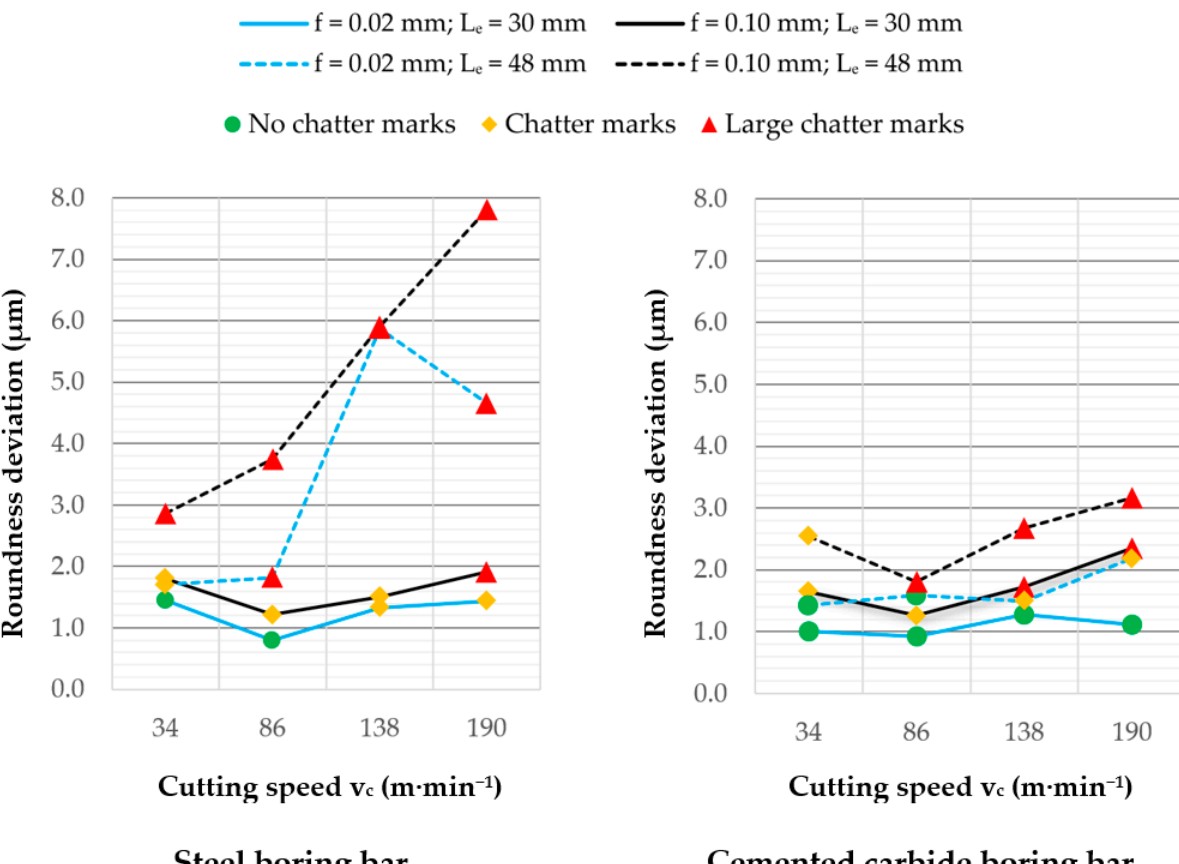

**Figure 12.** Average values of roundness.

However, in general, we can state that the dependence of roundness deviations on cutting speed values has a similar character to the measured surface roughness values in the graphs in Figures 7 and 8. This means that the same explanation can be used for roundness as mentioned for surface roughness in Section 3.1. In the case of the steel boring bar, internal turning with a tool overhang of 6D will be problematic with respect to roundness deviations if higher values of cutting speed are used, as can be seen in Figure 12. In this case, samples with cutting speed values of 138 and 190 m·min$^{-1}$ reached the largest roundness deviations. It means that if the cutting speed is increased, it will make the cutting process less stable, which is also indirectly reflected in larger roundness deviations. This proves that the previous phenomenon which was observed for the results of surface roughness of the machined surface can also be applied to this case. However, it is necessary to state that this phenomenon can be observed during turning with small diameters of the steel boring bar where unstable cutting conditions have occurred. In the case of the cemented carbide boring bar, samples with cutting speed values of 190 m·min$^{-1}$ reached the largest roundness deviation.

The roundness is not evaluated as often as surface roughness during the machining in scientific publications. Therefore, these results are particularly interesting as they can be considered to be one of the new contributions to this issue.

## 4. Conclusions

In the experiment, the influence of cutting speed on chatter marks, surface roughness and roundness was investigated. The effect of cutting speed was examined during the

various machining conditions where different values of feed, tool overhang and tool material of boring bar (steel and cemented carbide) were used.

The chatter marks or large chatter marks were observed as proof of unstable machining conditions during the experiment. The occurrence of chatter was directly related to the increase in the surface roughness parameters Ra and Rz of the machined surface. From the results, the cemented carbide boring bar can possibly be used for a specific range of cutting speeds where a sufficient quality of surface roughness (Ra < 0.8 μm, Rz < 4 μm and no chatter marks) can be achieved for internal turning with a tool overhang of 6D (6 × diameter of boring bar) or less in contrast to the steel boring bar. The steel boring bar is sufficient to be used for internal turning with a tool overhang of only 3.75D or less where a sufficient quality of surface roughness can be achieved. The lowest surface roughness parameters Ra and Rz were reached for a cutting speed of 86 m·min$^{-1}$, feed of 0.02 mm and tool overhang length of 30 mm for the steel as well as cemented carbide boring bar.

In the case of the steel boring bar, decreasing the cutting speed undoubtedly leads to an increase in the quality of the surface roughness and a reduction in the formation of chatter marks and large chatter marks. The cemented carbide boring bar also followed a similar trend, but it should be noted that the overall effect is not as significant. A reduction in the formation of chatter marks and large chatter marks was also observed. It can be stated from the results that decreasing the cutting speed leads to the improved surface quality of turned holes. However, it should be emphasized that this statement cannot be applied as a general phenomenon in machining. In fact, this phenomenon can be applied only to the conditions of internal turning by small-diameter boring bars where internal turning is critical due to unstable cutting conditions. The higher values of cutting speed make the cutting process less stable.

The results of roundness deviations appropriately complement the achieved results of surface roughness measurement. In general, it can be stated that the dependence of roundness deviations on cutting speed values has a similar character to the measured surface roughness values. It proved that the previous phenomenon which was observed for the results of surface roughness of the machined surface can also be applied to this case. These results are particularly interesting as they form one of the new contributions to this area of research.

The use of such low values of cutting speed is not at all typical when machining carbon steel materials with cemented carbide tools. On the other hand, decreasing the cutting speed makes the cutting process more stable, which results in a reduction in chatter mark formation. The authors assume that this would be related to lower values of tangential cutting force at a lower cutting speed. Therefore, further research will aim to prove this statement with an additional implementation of cutting force measurement. In addition, the adjustment of the cutting edge microgeometry in order to reduce the cutting force components will also be included in the subsequent research.

**Author Contributions:** Conceptualization, T.V.; data curation, T.V., B.B., A.G. and R.S.; formal analysis, T.V.; investigation, T.V.; methodology, T.V.; project administration, T.V.; resources, T.V., B.P., M.V. and F.J.; software, T.V., B.B. and A.G.; supervision, T.V.; validation, T.V.; visualization, T.V.; writing—original draft, T.V., B.B., B.P., M.V., F.J., M.K. and J.P.; writing—review and editing, T.V., B.B., B.P., M.V., F.J., M.K., J.P. and A.G. All authors have read and agreed to the published version of the manuscript.

**Funding:** This work was supported by the Slovak Research and Development Agency of the Slovak Republic under Contract No. APVV-21-0071 and VEGA grant agency of the Ministry of Education, Science, Research and Sport of the Slovak Republic and the Slovak Academy of Sciences No. 1/0266/23.

**Data Availability Statement:** The data presented in this study are available on request from the corresponding author. The data are not publicly available due to the large data capacity.

**Conflicts of Interest:** The authors declare no conflicts of interest.

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
