# Peer review of "The Selection of Cutting Speed to Prevent Deterioration of the Surface in Internal Turning of C45 Steel by Small-Diameter Boring Bars"

_machines, doi:10.3390/machines12010068_

Round 1

Reviewer 1 Report

Comments and Suggestions for Authors

The article deals with the experimental investigation of small diameter deep hole turning. Multiple tool materials, geometries and technoligical parameters were tested and evaluated. The topic fits the journal, section and special issue. Overall, the article is well-written, well-edited and contains some novel findings. The reviewer has the following comments:

1. Is the data unit in Table 1 is mm?

2. In Section 3.1, the stability limit is stated being 5D, just like in literature sources. How could the authors prove this conslusion as only tools with 3.75D and 6D were used? I have the same observation for page 10, line 258.

3. Although the the cutting speed is given throughout, the spindle speed might also be useful to mention.

4. It is described below Fig. 9 that there are some novel findings which are contradictory to earlier literature results. However, it is not clear to the reader what are the common parts and differences in the experiment(al setup) and other parameters between the current article and the ones cited. These should clearly be stated. The same is true for page 10, line 265.

Comments on the Quality of English Language

The English language level is sufficient. The work itself is understandable. Some typos, conjunctions can be found, though. A re-read is advised to eliminate these.

Author Response

Please see the attachment. The article together with the responses to the reviewer is at the end of the manuscript.

Reviewer 2 Report

Comments and Suggestions for Authors

In order to enhance the manuscript even further, my observations are as follows:

1. The citation of references in the manuscript should adhere to the template (for example, NOT: Consequently, researchers have directed their efforts towards improving machining parameters to enhance the process stability and optimize the turning performance when employing these small-diameter boring bars [6], [7], [8]. BUT: Consequently, researchers have directed their efforts towards improving machining parameters to enhance the process stability and optimize the turning performance when employing these small-diameter boring bars [6-8])." Correct throughout the entire paper.

2. References should be listed according to the template provided by the journal - reference [36].

3. The reference [34] is not cited.

4. Correct the DOIs in the references [15], [16], [21], [31].

5. Generally, improve the quality of all figures (align the font and font size to be consistent across all of them).

6. Figure 4 is too small, and the labels are not noticeable. Enlarge/improve the figure.

7. Figs. 10 and 11 are very unclear, and the legend is missing. Correct this.

8. The authors mention,”Experiments described in the paper deal with turning of C45 material, which is the 84 subject of machining and research by many authors [10], [11], [12] and others.” What did the authors [10], [11], [12] investigate, and what conclusions did they draw based on their research?

9. The authors mention,” “The tool overhang that is more than 4D - 5D is often considered as a critical value 97 [27], [17], [28], [29].” With respect to which quality criteria are the mentioned values critical (e.g., surface roughness, waviness, dimensional or geometric accuracy, etc.)? Do other variables (e.g., the geometry of the cutting tool, depth of cut, workpiece material, tool condition, etc.) influence these critical values?

10. If a statement is used, such as “It is well known from theory of machining that with in-127 creasing values of the depth of cut, the radial force of the tool on the workpiece also in-128 creases.” or “It is known from the literature revies that tool overhang that is 133 more than 4D - 5D is often considered as a critical value.” Please provide references supporting these statements.

11. I noticed that the authors used the term 'picture' in the paper (for example, Figure. 6 and text relate to the Fig.6). It is advisable to correct this and maintain consistency by using standard terms such as 'images,' 'figures,' or 'diagrams' for clarity and professionalism throughout the text.

12. Why in Figure 6, the third row of images, are there images with different cutting parameters shown, i.e., Vc=190m/min for the left image and Vc=138m/min for the right image? Please make the necessary corrections or adjustments.

13. After this sentence: “The machined surfaces of samples were sorted into three groups according to observed size of chatter marks: no chatter marks (marked by green circle), chatter marks (or ange rhombus), very large chatter marks (red triangle).” reference should be made to the images (Figure 7, Figure 8) as the entire paragraph appears unclear.

14. How would you recommend applying the obtained results in practice, especially in the industry?

15. Do you plan to conduct further research based on the obtained results? If so, what would be the main questions or areas of focus?

Comments on the Quality of English Language

Review the English language in the manuscript for accuracy and clarity.

Author Response

(The authors gave the same response as above.)
